# Accuracy of Accuhaler, Ellipta, and Turbuhaler Testers in Patients with Chronic Obstructive Pulmonary Disease

**DOI:** 10.3390/medsci13020050

**Published:** 2025-04-29

**Authors:** Narongkorn Saiphoklang, Thiravit Siriyothipun, Sarawut Panichaporn

**Affiliations:** Division of Pulmonary and Critical Care Medicine, Department of Internal Medicine, Faculty of Medicine, Thammasat University, Pathum Thani 12120, Thailand; thiravit.siri@gmail.com (T.S.); oui20081991@gmail.com (S.P.)

**Keywords:** Accuhaler, chronic obstructive pulmonary disease, Ellipta, peak inspiratory flow rate, Tubuhaler

## Abstract

**Background**: Peak inspiratory flow rate (PIFR) measurement is an essential tool for assessing the effectiveness of inhaler therapy in chronic obstructive pulmonary disease (COPD). This study aimed to evaluate the accuracy of three different inhaler testers compared to the In-Check DIAL^®^ device. **Methods**: A cross-sectional study was conducted in clinically stable COPD patients. Participants performed PIFR measurements using the In-Check DIAL^®^ device and three inhaler testers (Accuhaler, Ellipta, and Turbuhaler). Optimal PIFR was defined as ≥60 L/min. Minimum PIFR was defined as ≥30 L/min. **Results**: A total of 82 COPD patients (93.9% male) were included, with a mean age of 73.3 ± 8.8 years. Post-bronchodilator forced expiratory volume in one second was 69.2 ± 21.0%. The prevalence of optimal PIFR was 78%, 74%, and 52% for the Accuhaler, Ellipta, and Turbuhaler testers, respectively. For detecting optimal PIFR, the Accuhaler tester demonstrated an accuracy of 80.5%, sensitivity of 100%, and specificity of 11.1%. The Ellipta tester showed an accuracy of 78.1%, sensitivity of 100%, and specificity of 14.3%, while the Turbuhaler tester achieved an accuracy of 56.1%, sensitivity of 100%, and specificity of 7.7%. All devices showed excellent accuracy (>95%), sensitivity (>98%), and specificity (100% except for the Turbuhaler tester) in detecting minimum PIFR. **Conclusions**: The majority of COPD patients achieved optimal PIFR across the three different devices, with the highest prevalence observed for the Accuhaler tester. All three inhaler testers demonstrated excellent accuracy in assessing PIFR in COPD patients, suggesting their potential as reliable alternatives to the In-Check DIAL^®^ device in clinical practice.

## 1. Introduction

Chronic obstructive pulmonary disease (COPD) has emerged as a key global health problem. It ranks as the third leading cause of death globally, following ischemic heart disease and cerebrovascular disease [1]. While COPD remains incurable, symptoms can be managed and exacerbations reduced by not smoking, avoiding pollution, and using appropriate medication. Inhaled medications are the cornerstone of COPD treatment according to the Global Initiative for Chronic Obstructive Lung Disease (GOLD) [2]. The mainstay medications are long-acting beta-2 agonists (LABAs), long-acting muscarinic antagonists (LAMAs), and inhaled corticosteroids (ICSs). These medications are delivered through pressurized metered dose inhalers (pMDIs), soft mist inhalers (SMIs), or dry powder inhalers (DPIs). Device selection should consider not only the appropriate medication but also the patient’s ability to generate adequate inspiratory flow.

The optimal peak inspiratory flow rate (PIFR) is the maximal flow generated during a forced inspiratory maneuver, which is crucial for optimizing the DPI effectiveness in COPD patients [3,4]. Suboptimal PIFR (≤60 L/min) is common during acute exacerbation of COPD and predicts all-cause and COPD-related readmissions [5]. Patients with suboptimal PIFR who are discharged on nebulizers have significantly lower rates of COPD readmission compared to those discharged on DPIs [5].

Optimal PIFR requirements vary across different DPI devices. The Accuhaler tester requires a minimal flow of 30 L/min and operates optimally at 60 L/min, which significantly improves drug delivery and fine particle generation [4,6,7]. Similarly, the Turbuhaler tester requires a minimum of 30 L/min and performs optimally at 60 L/min, with drug delivery strongly correlating with flow rate [4,7,8]. The Ellipta tester, a medium-resistance device, delivers adequate drug output at standardized flow rates of ≥30 L/min and performs optimally at 60 L/min for both single and combination agents [4,9,10].

The In-Check DIAL^®^ device is considered the gold standard for measuring PIFR [11]. However, its use is limited due to a lack of familiarity and availability among general practitioners [12]. Selecting appropriate inhaler devices based on patients’ PIFR has the potential to improve treatment outcomes in obstructive airway diseases, especially when using more accessible testing devices [13]. Inhaler testers may serve as alternative tools for assessing inspiratory force for COPD patients. Therefore, the purpose of this study was to evaluate the accuracy of three inhaler testers—Accuhaler, Ellipta, and Turbuhaler—compared to the In-Check DIAL^®^ device in COPD patients.

## 2. Materials and Methods

### 2.1. Study Design and Participants

Between March 2024 and December 2024, a cross-sectional study was undertaken at the pulmonary outpatient department of Thammasat University Hospital in Thailand. The inclusion criteria were (1) patients aged 40 years or older; (2) a smoking history of 10 pack-years or more; and (3) a diagnosis of COPD confirmed by a post-bronchodilator (BD) forced expiratory volume in one second (FEV_1_) to forced vital capacity (FVC) ratio of less than 0.7. The exclusion criteria were (1) COPD exacerbation within 3 months prior to study recruitment; (2) the presence of other pulmonary diseases, such as asthma, bronchiectasis, or pulmonary fibrosis; (3) a history of stroke with upper limb weakness or paresis; (4) any conditions or medications causing muscle weakness; (5) inability to perform testing with inhaler testers or In-Check DIAL^®^; (6) tracheostomy or the need for home ventilator support (both invasive and non-invasive); and (7) inability to communication or follow to instructions.

Ethics approval was obtained from the Human Research Ethics Committee of Thammasat University (Medicine), Thailand (IRB No. MTU-EC-IM-0-016/67, COA No.095/2024, date of approval: 28 March 2024), in full compliance with international guidelines, including the Declaration of Helsinki, the Belmont Report, CIOMS Guidelines, and the International Conference on Harmonization Good Clinical Practice (ICH-GCP). All methods were performed in accordance with these guidelines and regulations. Written informed consent was obtained from all participants. This study was registered on ClinicalTrials.gov with the number NCT06346678.

### 2.2. Study Procedures

Demographic data, respiratory symptoms, and functional capacity (assessed using the modified Medical Research Council (mMRC) dyspnea scale [14] and the COPD Assessment Test (CAT) [15]), as well as spirometry data from the past 12 months, were collected. Baseline medications, including short-acting bronchodilator (SABD), ICS, LABA, and LAMA, were also recorded.

The severity of COPD, according to the GOLD classification, was determined using the post-BD FEV_1_ value: Grade 1 represented mild (≥80% of predicted value); Grade 2 was moderate (50–79%); and Grades 3 and 4 represented severe (<50%) and very severe (<30%) impairment, respectively [2]. Based on symptom burden and exacerbation history, patients were categorized into Groups A, B, and E [2].

PIFR was measured using the In-Check DIAL^®^ device, as well as the Accuhaler, Ellipta, and Turbuhaler testers. Each device was tested three times with one-minute intervals between the tests, and the highest value was recorded. The testing sequence was randomized according to six different orders: (1) Accuhaler–Ellipta–Turbuhaler; (2) Turbuhaler–Ellipta–Accuhaler; (3) Ellipta–Accuhaler–Turbuhaler; (4) Ellipta–Turbuhaler–Accuhaler; (5) Turbuhaler–Accuhaler–Ellipta; or (6) Accuhaler–Turbuhaler–Ellipta. For each sequence, the In-Check DIAL^®^ resistance was adjusted to match the corresponding tester device before testing with that device.

### 2.3. Outcomes

The primary outcomes were the accuracy, sensitivity, and specificity of the three inhaler testers in identifying optimal PIFR compared to the In-Check DIAL^®^ device. The secondary outcomes included the prevalence rates of optimal, suboptimal, minimum, and insufficient PIFR. Additionally, factors associated with suboptimal PIFR were also considered secondary outcomes.

PIFR classifications were based on PIFR values [4,7,16,17]: optimal PIFR (≥60 L/min), suboptimal PIFR (<60 L/min), minimum PIFR (≥30 L/min), and insufficient PIFR (<30 L/min).

### 2.4. Statistical Analysis

The accuracy of inhaler testers in COPD patients has not been investigated. A study by Manuyakorn W et al. [18] reported the Accuhaler tester having a sensitivity of 95.4% in adolescents with asthma. We hypothesized that the sensitivity of the Accuhaler tester in COPD patients would be 85%. A sample size of 80 was proposed to achieve an alpha of 0.03 and a power of 0.86.

Descriptive statistics are presented as numbers (%) and mean ± standard deviation. Sensitivity, specificity, positive predictive value (PPV), negative predictive value (NPV), and accuracy were reported. The chi-squared test was used to compare categorical variables between the optimal and suboptimal PIFR groups. The independent *t*-test or Mann–Whitney U test was used to compare continuous variables between the two groups. Statistical analyses were conducted using SPSS software (version 25.0; IBM Corp., Armonk, NY, USA), and a two-sided *p*-value of <0.05 was considered statistically significant.

## 3. Results

### 3.1. Participants

Eighty-two COPD patients (93.9% male) were included, with a mean age of 73.3 ± 8.8 years. Common comorbidities included hypertension (61.0%), dyslipidemia (47.6%), and diabetes mellitus (20.7%). COPD Grade 2 and a higher proportion of Group E were commonly observed (40.2% and 40.3%, respectively). Triple inhalation therapy (ICS/LABA/LAMA) was the most frequent maintenance treatment (48.8%). The CAT scores were 9.1 ± 5.7, and the mMRC scores were 1.5 ± 1.1. Post-BD FEV_1_ was 69.2 ± 21.0% (Table 1).

### 3.2. Primary Outcomes

For detecting optimal PIFR, the Accuhaler, Ellipta, and Turbuhaler testers demonstrated accuracies of 80.5%, 78.1%, and 56.1%, respectively. All inhaler testers exhibited 100% sensitivity but low specificity (11.1%, 14.3%, and 7.7% for the Accuhaler, Ellipta, and Turbuhaler testers, respectively) (Table 2 and Figure 1). However, the accuracy and specificity of the Accuhaler, Ellipta, and Turbuhaler testers were higher when detecting minimum PIFR (Table 2).

### 3.3. Secondary Outcomes

The prevalence rates of optimal, suboptimal, minimum, and insufficient PIFR were as follows: for Accuhaler, 78.0%, 20.7%, 98.8%, and 1.2%, respectively; for Ellipta, 74.4%, 25.6%, 97.6%, and 2.4%; and for Turbuhaler, 52.4%, 47.6%, 93.9%, and 6.1%, respectively (Table 3).

For the Accuhaler tester, the factors associated with suboptimal PIFR included older age, lower body weight, a higher proportion of coronary artery disease, and higher CAT and mMRC scores. For the Ellipta tester, factors included older age, lower body weight and height, a higher amount of smoking, a higher proportion of atrial fibrillation, higher CAT and mMRC scores, a higher proportion of GOLD Group E, and a higher proportion of SABD use. For the Turbuhaler tester, factors included older age, lower body weight, height, and body mass index, lower FVC, higher CAT and mMRC scores, a higher proportion of GOLD Group E, and a higher proportion of SABD use (Table 4).

## 4. Discussion

This is the first study to evaluate three inhaler testers compared to the In-Check DIAL^®^ device for PIFR measurement in COPD patients. Our findings revealed that all testers exhibited very high sensitivity and NPV (100%) but low specificity for optimal PIFR (7.7–14.3%). The accuracy of the Accuhaler and Ellipta testers (80.5% and 78.1%, respectively) was superior to that of the Turbuhaler tester (56.1%). The high sensitivity and NPV indicate that all the testers can be used for selecting an appropriate DPI for COPD patients. Although their low specificity indicates a tendency to produce false positives, the testers remain useful for minimizing inappropriate exclusion of patients from DPI therapy. Overall, the testers’ high sensitivity but low specificity suggests they are better suited as screening tools than replacements for the In-Check DIAL^®^ device.

Minimum PIFR detection (≥30 L/min) for all the inhaler testers is a remarkable finding in our study. The Accuhaler and Ellipta testers demonstrated identical superior accuracy (98.8%) with perfect specificity (100%) and excellent sensitivity (98.8%). With strong PPVs (100%), these testers can effectively identify patients capable of using a DPI device. The Turbuhaler tester also showed excellent accuracy (95.1%) and high sensitivity (98.7%) despite having low specificity (40.0%). These minimum PIFR detection outcomes are beneficial for selecting DPI devices to deliver inhaled medication to the lungs, thereby improving treatment effectiveness. Based on these findings, these testers may be most useful for identifying patients with insufficient inspiratory force, although their PIFR may not necessarily be optimal.

Interestingly, our findings in the COPD study correspond to those of Manuyakorn W et al. [18] in asthmatic children and adolescents, despite differences in disease pathophysiology and patient age. The Accuhaler and Turbuhaler testers in their study showed slightly lower sensitivity than ours for identifying optimal PIFR (97% vs. 100% for Accuhaler and 98% vs. 100% for Turbuhaler). These findings suggest that the Accuhaler and Turbuhaler testers can be effectively used across patients with different baseline diseases and characteristics. However, the detection of suboptimal PIFR in our COPD patients (22.0% and 47.6% for Accuhaler and Turbuhaler, respectively) was significantly higher than in asthmatic children and adolescents (0% and 0–10% for Accuhaler and Turbuhaler, respectively). This highlights the importance of measuring PIFR before selecting DPIs in COPD patients.

A study by Melani AS et al. [19], which involved 644 patients, including those with asthma and COPD, assessed PIFR using the Diskus (Accuhaler) inhaler with the In-Check DIAL^®^ device. It was found that 60% of patients with initially weak inhalation efforts had a PIFR below 30 L/min. However, after a brief instructional session emphasizing the need for more forceful inhalation, all patients achieved a PIFR of at least 30 L/min, indicating that proper technique can significantly improve inhaler performance. In contrast, when using the Turbuhaler tester, 77% demonstrated a PIF < 30 L/min. After counseling, 12% of patients still did not achieve a PIFR of at least 30 L/min.

In a study of 101 adult asthma patients by Engel T et al. [20], PIFR was measured both with and without the Turbuhaler device. While PIFR using the Turbuhaler tester was significantly lower than without it, only 4% of patients had a PIFR below 30 L/min, which is considered the minimum for effective drug delivery. This suggests that most patients can generate sufficient inspiratory flow using Turbuhaler. Another study by Brown P.H. assessed PIFR in 99 adults presenting with acute asthma exacerbations [21]. It was found that 98% of patients achieved a PIFR of at least 30 L/min using the Turbuhaler device, even before bronchodilator treatment, indicating that the majority could effectively use the device during acute episodes.

A randomized cross-over trial by Altman P et al. [22] compared PIFR among COPD patients using the Ellipta, Breezhaler, and HandiHaler devices. The study found that the mean PIFR achieved with the Ellipta inhaler was 78 L/min, which was higher than that with HandiHaler (49 L/min) but lower than with Breezhaler (108 L/min). This suggests that the Ellipta inhaler requires a moderate level of inspiratory effort, making it suitable for many COPD patients. These studies underscore the importance of assessing inspiratory flow rates when selecting an appropriate inhaler device for patients, as well as the potential benefits of patient education on inhaler technique to ensure effective drug delivery. Based on our study findings, if a patient’s test result is positive using the Accuhaler or Ellipta tester, it can be reasonably assumed that the patient has an optimal PIFR, as both devices demonstrated relatively high PPVs (80% and 77.2%, respectively). In contrast, the Turbuhaler tester showed a PPV of 54.4%, indicating a higher likelihood of false-positive results. In such cases, the In-Check DIAL^®^ device is needed to confirm optimal PIFR. However, if testing with any of these testers yields negative results, it can be reliably concluded that the patient cannot generate a flow of at least 60 L/min for optimal PIFR, as all three testers demonstrated strong NPVs.

Our study found that 22%, 25.6%, and 47.6% of patients had suboptimal PIFR using the Accuhaler, Ellipta, and Turbuhaler testers, respectively. These findings are consistent with previous studies, which reported suboptimal PIFR ranging from 20.1% to 78% [5,23,24,25,26,27]. Insufficient PIFR (<30 L/min) was identified in only 1.2% to 6.1% of patients in our study, indicating that most stable COPD patients can generate the minimum required inspiratory flow for DPI use. These results support the use of DPI devices in COPD therapy. Therefore, if treatment effectiveness remains inadequate during DPI use, inspiratory flow testing should be performed to evaluate whether the device is suitable for the patient.

The factors associated with suboptimal PIFR in our COPD patients included older age, lower body weight, height, and body mass index, a higher smoking history, higher proportions of coronary artery disease and atrial fibrillation, and higher CAT and mMRC scores. Other factors included a higher proportion of GOLD Group E, lower FVC, and a higher proportion of SABD use. In a study by Suriyakul A et al. [26], hand grip strength, age, height, and FVC were identified as predictors for Accuhaler PIFR, while hand grip strength, female gender, age, and FVC were predictors for Turbuhaler PIFR in COPD patients. Represas-Represas C et al. [23] found that age and FVC were significantly associated with suboptimal PIFR in stable COPD patients. Additionally, a study by Duarte AG et al. [28] showed that PIFR correlated with inspiratory capacity (r = 0.40, *p* < 0.0001) and the ratio of residual volume to total lung capacity (r = −0.19, *p* = 0.002), indicating that air trapping impacts PIFR in COPD patients. Our study suggests that older age, lower body mass index, higher respiratory symptoms, a history of COPD exacerbation, frequent rescue SABD use, presence of heart disease, and lower lung function were associated with lower PIFR values. These predictors could be useful for physicians when selecting the appropriate inhaler devices for individual patients. They also suggest that physicians should consider measuring PIFR before prescribing medications with DPI devices to maximize drug delivery.

This study has a few limitations. Firstly, the findings might not be applicable to the broader population of individuals with COPD, as this was a single-center study that excluded patients with recent exacerbation or significant comorbidities. Additionally, the sample was predominantly male (94%), and potential order effects—such as learning or fatigue—may have influenced the outcomes. Secondly, although the testing sequence was randomized to minimize assessment bias, patient fatigue and learning effects may have influenced the results. Multicenter studies are needed to validate the inhaler tests in heterogeneous COPD cohorts and evaluate PIFR-guided device selection.

## 5. Conclusions

The majority of COPD patients achieved optimal PIFR across different devices, with the Accuhaler tester showing the highest prevalence. Several factors were associated with suboptimal PIFR. All three inhaler testers demonstrated excellent accuracy in assessing PIFR in COPD patients, indicating their potential as reliable alternatives to the In-Check DIAL^®^ device in clinical practice. However, the testers’ high sensitivity but low specificity suggests they are better suited as screening tools than replacements for the In-Check DIAL^®^ device. These findings suggest that these devices could be effectively integrated into routine clinical assessments for managing COPD.

## Figures and Tables

**Figure 1 medsci-13-00050-f001:**
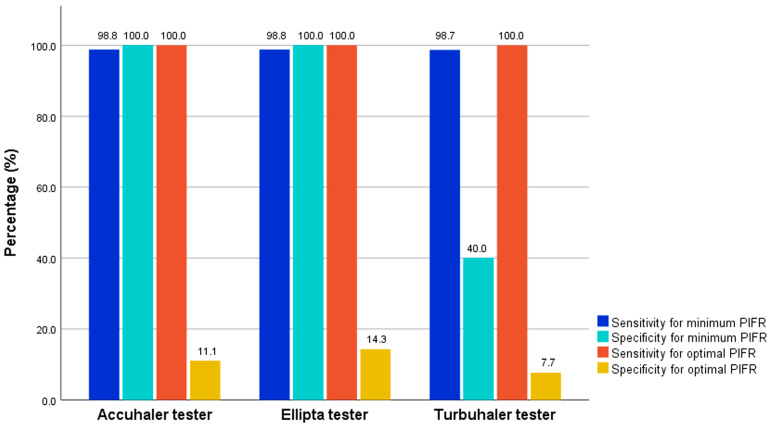
Sensitivity and specificity of testers for detecting minimum and optimal PIFR in COPD patients. COPD = chronic obstructive pulmonary disease, PIFR = peak inspiratory flow rate.

**Table 1 medsci-13-00050-t001:** Baseline characteristics of COPD patients.

Characteristics	Data (n = 82)
Age, years	73.3 ± 8.8
Male/female	77 (93.9)/5 (6.1)
Body mass index, kg/m^2^	22.0 ± 3.9
Smoking, pack-years	30.6 ± 23.4
Comorbidity	
Hypertension	50 (61.0)
Dyslipidemia	39 (47.6)
Diabetes mellitus	17 (20.7)
Coronary artery disease	13 (15.9)
Atrial fibrillation	5 (6.1)
Congestive heart failure	2 (2.4)
Obstructive sleep apnea	5 (6.1)
Allergic rhinitis	6 (7.3)
COPD Grade	
1	31 (37.8)
2	33 (40.2)
3	15 (18.3)
4	3 (3.7)
COPD Group	
A	32 (39.0)
B	17 (20.7)
E	33 (40.3)
Medication	
SABD	38 (46.3)
LAMA	7 (8.5)
LABA/LAMA	28 (34.1)
ICS/LABA	7 (8.5)
ICS/LABA/LAMA	40 (48.8)
Methylxanthine	15 (18.3)
Oral beta-2 agonist	3 (3.7)
Macrolide	2 (2.4)
PDE4 inhibitor	2 (2.4)
Current inhalation device	
Metered dose inhaler	11 (13.4)
Accuhaler	8 (9.8)
Turbuhaler	6 (7.3)
Ellipta	36 (43.9)
Soft mist inhaler	23 (28.0)
HandiHaler	16 (19.5)
Functional capacity	
CAT scores	9.1 ± 5.7
mMRC scores	1.5 ± 1.1
Spirometry data	
Post-bronchodilator FVC, L	2.93 ± 0.82
Post-bronchodilator FVC, %predicted	94.4 ± 19.4
Post-bronchodilator FEV_1_, L	1.62 ± 0.58
Post-bronchodilator FEV_1_, %predicted	69.2 ± 21.0
Post bronchodilator FEV_1_/FVC, %	54.9 ± 11.9

Data are presented as n (%) or mean ± SD. CAT = COPD Assessment Test, COPD = chronic obstructive pulmonary disease, mMRC = modified Medical Research Council, FEV_1_ = forced expiratory volume in one second, FVC = forced vital capacity, ICS = inhaled corticosteroids, kg = kilograms, L = liters, LABA = long-acting beta-2 agonist, LAMA = long-acting muscarinic antagonist, m = meter, mm = millimeter, PDE4 = phosphodiesterase-4, SABD = short-acting bronchodilator.

**Table 2 medsci-13-00050-t002:** Sensitivity and specificity of testers for detecting minimum and optimal PIFR in COPD patients.

Parameter	Accuhaler Tester	Ellipta Tester	Turbuhaler Tester
Minimum PIFR			
Sensitivity, %	98.8	98.8	98.7
Specificity, %	100.0	100.0	40.0
PPV, %	100.0	100.0	96.2
NPV, %	50.0	66.7	66.7
Accuracy, %	98.8	98.8	95.1
Optimal PIFR			
Sensitivity, %	100.0	100.0	100.0
Specificity, %	11.1	14.3	7.7
PPV, %	80.0	77.2	54.4
NPV, %	100.0	100.0	100.0
Accuracy, %	80.5	78.1	56.1

COPD = chronic obstructive pulmonary disease, L = liter, NPV = negative predictive value, PIFR = peak inspiratory flow rate, PPV = positive predictive value. Minimum PIFR was defined as PIFR ≥ 30 L/min; optimal PIFR was defined as PIFR ≥ 60 L/min.

**Table 3 medsci-13-00050-t003:** Peak inspiratory flow rate in COPD patients.

Parameter	Accuhaler (n = 82)	Ellipta (n = 82)	Turbuhaler (n = 82)
PIFR, L/min	71.5 ± 19.0	70.8 ± 18.3	59.0 ± 17.2
Optimal PIFR	64 (78.0)	61 (74.4)	43 (52.4)
Suboptimal PIFR	18 (22.0)	21 (25.6)	39 (47.6)
Minimum PIFR	81 (98.8)	80 (97.6)	77 (93.9)
Insufficient PIFR	1 (1.2)	2 (2.4)	5 (6.1)

Data are presented as n (%) or mean ± SD. COPD = chronic obstructive pulmonary disease, L = liter, PIFR = peak inspiratory flow rate. Optimal PIFR was defined as PIFR ≥ 60 L/min, suboptimal PIFR was defined as PIFR < 60 L/min, minimum PIFR was defined as PIFR ≥ 30 L/min, and insufficient PIFR was defined as PIFR < 30 L/min.

**Table 4 medsci-13-00050-t004:** Factors associated with optimal PIFR in COPD patients for Accuhaler, Ellipta, and Turbuhaler.

Variable	Accuhaler (n = 82)			Ellipta (n = 82)			Turbuhaler (n = 82)		
	Optimal	Suboptimal	*p*-Value	Optimal	Suboptimal	*p*-Value	Optimal	Suboptimal	*p*-Value
Patients	64 (78.0)	18 (22.0)	NA	61 (74.4)	21 (25.6)	NA	43 (52.4)	39 (47.6)	NA
Maximal PIFR, L/min	78.8 ± 14.0	45.5 ± 9.0	<0.001	78.7 ± 13.0	47.9 ± 10.0	<0.001	72.5 ± 9.0	44.2 ± 10.3	<0.001
Sex			0.068			0.103			0.186
Male	62 (96.9)	15 (83.3)		59 (96.7)	18 (85.7)		42 (97.7)	35 (89.7)	
Female	2 (3.1)	3 (16.7)		2 (3.3)	3 (14.3)		1 (2.3)	4 (10.3)	
Age, years	72.3 ± 8.9	77.0 ± 7.8	0.043	71.7 ± 8.8	77.8 ± 7.5	0.006	69.7 ± 8.9	77.2 ± 6.9	<0.001
Body weight, kg	61.9 ± 12.4	55.2 ± 13.1	0.049	62.1 ± 11.8	55.4 ± 14.3	0.037	64.4 ± 11.5	56.0 ± 12.8	0.003
Height, cm	166.1 ± 6.9	163.0 ± 9.3	0.124	166.6 ± 6.8	162.0 ± 8.7	0.015	167.7 ± 6.6	162.9 ± 7.8	0.004
BMI, kg/m^2^	22.3 ± 3.8	20.7 ± 4.0	0.106	22.3 ± 3.4	21.0 ± 4.9	0.202	22.8 ± 3.4	21.0 ± 4.1	0.030
Active smoking	6 (9.4)	1 (5.6)	0.372	6 (9.8)	1 (4.8)	0.219	5 (11.6)	2 (5.1)	0.276
Smoking, pack-years	28.5 ± 24.5	38.3 ± 17.7	0.119	25.7 ± 19.0	44.9 ± 29.1	0.001	26.8 ± 18.7	34.9 ± 27.3	0.116
Comorbidity									
Hypertension	39 (60.9)	11 (61.1)	0.989	38 (62.3)	12 (57.1)	0.676	25 (58.1)	25 (64.1)	0.580
Dyslipidemia	30 (46.9)	9 (50.0)	0.815	27 (44.3)	12 (57.1)	0.308	19 (44.2)	20 (51.3)	0.521
Diabetes mellitus	13 (20.3)	4 (22.2)	1.000	13 (21.3)	4 (19.0)	1.000	10 (23.3)	7 (17.9)	0.554
Coronary artery disease	7 (10.9)	6 (33.3)	0.032	8 (13.1)	5 (23.8)	0.302	6 (14.0)	7 (17.9)	0.621
Atrial fibrillation	2 (3.1)	3 (16.7)	0.068	1 (1.6)	4 (19.0)	0.014	1 (2.3)	4 (10.3)	0.186
Congestive heart failure	2 (3.1)	0 (0)	1.000	2 (3.3)	0 (0)	1.000	1 (2.3)	1 (2.6)	1.000
Obstructive sleep apnea	4 (6.3)	1 (5.6)	1.000	4 (6.6)	1 (4.8)	1.000	3 (7.0)	2 (5.1)	1.000
Allergic rhinitis	6 (9.4)	0 (0)	0.330	6 (9.8)	0 (0)	0.330	1 (2.3)	5 (12.8)	0.097
Spirometry data									
Post-BD FEV_1_, %	70.4 ± 21.1	64.9 ± 20.4	0.331	71.0 ± 20.5	63.8 ± 22.0	0.181	72.9 ± 19.4	65.0 ± 22.2	0.092
Post-BD FVC, %	99.4 ± 19.9	90.3 ± 16.3	0.080	99.3 ± 20.1	91.8 ± 16.7	0.128	101.5 ± 18.7	92.9 ± 19.4	0.046
COPD Grade 3 and 4	13 (20.3)	5 (27.8)	0.527	12 (19.7)	6 (28.6)	0.541	8 (18.6)	10 (25.6)	0.442
Functional performance									
CAT scores	8.3 ± 4.8	12.0 ± 7.8	0.014	7.9 ± 4.8	12.4 ± 7.1	0.013	7.2 ± 4.2	11.2 ± 6.5	0.002
CAT ≥ 10	24 (37.5)	11 (61.1)	0.074	22 (36.1)	13 (61.9)	0.039	13 (30.2)	22 (56.4)	0.017
mMRC scores	1.3 ± 1.1	2.3 ± 1.0	0.001	1.2 ± 1.0	2.4 ± 1.0	<0.001	1.1 ± 0.9	2.0 ± 1.2	<0.001
mMRC ≥ 2	19 (29.7)	13 (72.2)	0.001	16 (26.2)	16 (76.2)	<0.001	9 (20.9)	23 (59.0)	<0.001
GOLD Group E	24 (37.5)	9 (50.0)	0.339	20 (32.8)	13 (61.9)	0.019	11 (25.6)	22 (56.4)	0.004
Medication									
SABD	27 (42.2)	11 (61.1)	0.155	24 (39.3)	14 (66.7)	0.030	14 (32.6)	24 (61.5)	0.009
LAMA	4 (6.3)	3 (16.7)	0.175	4 (6.6)	3 (14.3)	0.365	4 (9.3)	3 (7.7)	1.000
LABA/LAMA	21 (32.8)	7 (38.9)	0.631	22 (36.1)	6 (28.6)	0.532	15 (34.9)	13 (33.3)	0.882
ICS/LABA	7 (10.9)	0 (0)	0.338	7 (11.5)	0 (0)	0.182	5 (11.6)	2 (5.1)	0.436
ICS/LABA/LAMA	32 (50.0)	8 (44.4)	0.677	28 (45.9)	12 (57.1)	0.374	19 (44.2)	21 (53.8)	0.382
Methylxanthine	11 (17.2)	4 (22.2)	0.731	10 (16.4)	5 (23.8)	0.516	6 (14.0)	9 (23.1)	0.286
Oral beta-2 agonist	3 (4.7)	0 (0)	1.000	3 (4.9)	0 (0)	0.566	2 (4.7)	1 (2.6)	1.000
Macrolide	1 (1.6)	1 (5.6)	0.393	1 (1.6)	1 (4.8)	0.449	0 (0)	2 (5.1)	0.223
PDE4 inhibitor	2 (3.1)	0 (0)	1.000	1 (1.6)	1 (4.8)	0.449	0 (0)	2 (5.1)	0.223

Data are presented n (%) or mean ± SD. BD = bronchodilator, BMI = body mass index, CAT = COPD Assessment Test, DBP = diastolic blood pressure, FEV_1_ = force expiratory volume in 1 s, FVC = forced vital capacity, GOLD = Global Initiative for Obstructive Lung Disease, HGS = hand grip strength, ICS = inhaled corticosteroid, LABA = long-acting beta-2 agonist, LAMA = long-acting muscarinic antagonist, mMRC = modified Medical Research Council, PDE4 = phosphodiesterase-4, PIFR = peak inspiratory flow rate, SABD = short-acting bronchodilator, SBP = systolic blood pressure, SpO2 = oxygen saturation.

## Data Availability

The data supporting the results of this study are available within the article.

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
