# Peer review of "Accuracy of Accuhaler, Ellipta, and Turbuhaler Testers in Patients with Chronic Obstructive Pulmonary Disease"

_medsci, 2025, doi:10.3390/medsci13020050_

Round 1
Reviewer 1 Report
Comments and Suggestions for Authors
Please limit the title to the study population. The authors examined data from a single center in Thailand.
Data from Table 1 should include information about the unit, e.g. Hypertension (%); Smoking, pack-years (mean±SD), etc. The data will be easier to read. Or, for each differentiating data, place an appropriate descriptor: * or #.
The discussion should be supplemented with other screening studies conducted in other populations.
Author Response
Comments 1: Please limit the title to the study population. The authors examined data from a single center in Thailand.
Response 1: We would like to express my heartfelt gratitude to the reviewer for the wonderful reviews and comments. Thank you very much for your suggestion regarding the study title. However, we are unable to change the title due to ethical and funding constraints.
Comments 2: Data from Table 1 should include information about the unit, e.g. Hypertension (%); Smoking, pack-years (mean±SD), etc. The data will be easier to read. Or, for each differentiating data, place an appropriate descriptor: * or #.
Response 2: Thank you very much for your suggestion regarding information about the unit. However, we have already presented the data as n (%) or mean±SD in the footnotes of all tables.
Comments 3: The discussion should be supplemented with other screening studies conducted in other populations.
Response 3: We have added a discussion of other screening studies in the Discussion section (pages 10-11, lines 224–249). Thank you very much.

Reviewer 2 Report
Comments and Suggestions for Authors
Dear Authors,
Thank you for the opportunity to review your manuscript. This is an excellent and clinically relevant study assessing the accuracy of widely used dry powder inhaler testers to quantify PIFR in patients with COPD. The use of In-Check DIAL as a reference standard, along with clear methodology and full statistical analysis, strengthens the study.
I have only a few minor suggestions to improve the manuscript's clarity and impact:
1. Although testers were very sensitive, their low specificity means they would be most useful as a screening tool, rather than a substitute for the In-Check DIAL. Clarify this more in the discussion and conclusions.
2. Refer to potential order effects (e.g., learning or fatigue), and highlight the single center setting and male-heavy sample as limitations to generalizability.
3. An example (bar chart) of sensitivity and specificity between devices can make it easier to read.
4. Make clear that these testers are of most value to identify patients with an improbable but not necessarily optimal PIFR.
5. Encourage studies that identify whether using these testers is linked to an improvement in inhaler choice, compliance, or outcomes.
In total, I recommend acceptance after revisions. It is a good contribution to the field.
Yours sincerely
Author Response
Comments: Thank you for the opportunity to review your manuscript. This is an excellent and clinically relevant study assessing the accuracy of widely used dry powder inhaler testers to quantify PIFR in patients with COPD. The use of In-Check DIAL as a reference standard, along with clear methodology and full statistical analysis, strengthens the study.
I have only a few minor suggestions to improve the manuscript's clarity and impact:
Response: I would like to express my heartfelt gratitude to the reviewer for the wonderful reviews and comments. I will do my best.
Comments 1: Although testers were very sensitive, their low specificity means they would be most useful as a screening tool, rather than a substitute for the In-Check DIAL. Clarify this more in the discussion and conclusions.
Response 1: Thank you for your valuable suggestion. We have addressed this point in the Discussion section (page 10, lines 200–202) and the Conclusions section (page 12, lines 299–301).
Comments 2: Refer to potential order effects (e.g., learning or fatigue), and highlight the single center setting and male-heavy sample as limitations to generalizability.
Response 2: Thank you for your insightful comment. We have addressed this point in the Discussion section (page 12, lines 285–289).
Comments 3: An example (bar chart) of sensitivity and specificity between devices can make it easier to read.
Response 3: We have added this figure in the Results section (page 6, Figure 1). Thank you for the suggestion.
Comments 4: Make clear that these testers are of most value to identify patients with an improbable but not necessarily optimal PIFR.
Response 4: We have addressed this point in the Discussion section (page 10, lines 210–212).
Comments 5: Encourage studies that identify whether using these testers is linked to an improvement in inhaler choice, compliance, or outcomes.
Response 5: We have included relevant studies in the Discussion section (pages 10-11, lines 224–249). Thank you very much.
